# Carrier-Free Hybrid Nanoparticles for Enhanced Photodynamic Therapy in Oral Carcinoma via Reversal of Hypoxia and Oxidative Resistance

**DOI:** 10.3390/pharmaceutics16091130

**Published:** 2024-08-27

**Authors:** Xiao Li, Zhiyin Li, Yue Su, Jia Zhou, Yuxiang Li, Qianqian Zhao, Xia Yang, Leilei Shi, Lingyue Shen

**Affiliations:** 1Department of Cleft Palate Speech, Department of Oral and Maxillofacial Surgery, The First Affiliated Hospital of Harbin Medical University, Harbin 150001, China; lixiao@hrbmu.edu.cn; 2Department of Oral and Maxillofacial-Head and Neck Oncology, Department of Laser and Aesthetic Medicine, Shanghai Ninth People’s Hospital, Shanghai Jiao Tong University School of Medicine, College of Stomatology, Shanghai Jiao Tong University, National Center for Stomatology, National Clinical Research Center for Oral Diseases, Shanghai Key Laboratory of Stomatology, Shanghai Research Institute of Stomatology, Shanghai Center of Head and Neck Oncology Clinical and Translational Science, Shanghai 200011, China; lizhiyin-1116-sjtu@sjtu.edu.cn (Z.L.); yangx2257@sh9hospital.org.cn (X.Y.); 3School of Chemistry and Chemical Engineering, Frontiers Science Center for Transformative Molecules, Shanghai Jiao Tong University, 800 Dongchuan Road, Shanghai 200240, China; yuesu@sjtu.edu.cn (Y.S.); lyx13062000501@sjtu.edu.cn (Y.L.); 4Department of Radiology, Shanghai Sixth People’s Hospital Affiliated to Shang Hai Jiao Tong University School of Medicine, Shanghai 200025, China; jia0531@sjtu.edu.cn (J.Z.); zhaoqianq5@sjtu.edu.cn (Q.Z.); 5Precision Research Center for Refractory Diseases in Shanghai General Hospital, Shanghai Jiao Tong University School of Medicine, Shanghai 200025, China

**Keywords:** coordinate self-assembled nanoparticles, HIF-1α inhibitor, Fenton reaction, photodynamic therapy

## Abstract

In the present work, we pioneered a coordinated self-assembly approach aimed at fabricating carrier-free hybrid nanoparticles to address the inherent challenges of the anaerobic microenvironment and the oxidative resistance induced by reductive glutathione (GSH) in photodynamic therapy (PDT). In these nanoparticles, protoporphyrin IX (PP), HIF-1α inhibitor of *N, N*ʹ-(2,5-Dichlorosulfonyl) cystamine KC7F2 (KC), and the cofactor Fe^3+^ present hydrogen bond and coordination interaction. The nanoparticles exhibited efficient cellular uptake by CAL-27 cells, facilitating their accumulation in tumors by enhanced permeability and retention (EPR) effect. Under irradiation at 650 nm, the formation of cytotoxic singlet oxygen (^1^O_2_) would be enhanced by the synergy effect on the Fenton reaction of Fe^3+^ ion and the downregulation of the HIF-1α, leading to the improved PDT efficacy both in vitro and in vivo biological studies. Our work opens a new supramolecular approach to prepare hybrid nanoparticles for effective synergy therapy with PDT against cancer cells.

## 1. Introduction

Oral carcinoma has been recognized as a global and prominent malignant neoplasm, with an estimated annual incidence of 355,000 new cases. Notably, over 90% of these cases are classified as squamous cell carcinomas in regard to histopathology, predominantly occurring in areas of superficial tissue covered by squamous epithelium [1,2]. Traditionally, the therapeutic approach to oral carcinoma has combined surgical excision with comprehensive chemoradiation [3,4]. However, a significant number of these patients undergoing this regimen experience adverse outcomes such as tumor recurrence, development of drug-resistant phenotypes, and metastatic progression. Additionally, post-treatment quality of life often deteriorates due to psychosocial impacts and functional impairments, including tissue morbidity, xerostomia, mucositis, and fibrotic changes [5,6,7,8]. These challenges underscore the necessity to refine and optimize therapeutic strategies, particularly for patients with advanced-stage oral carcinoma or extensive superficial squamous lesions [9,10,11]. In recent years, PDT has emerged as a leading modality for treating oral carcinoma. PDT offers significant advantages in managing superficial tumors due to its remarkable specificity, minimally invasive nature, repeatability, pain reduction, rapid therapeutic effects, minimal side effects, and swift systemic clearance [12,13,14,15]. Thus, the judicious application of PDT in patients with widespread oral squamous cell carcinoma (OSCC) contributes to preserving oral functions [16]. Especially for individuals with advanced-stage OSCC, PDT represents a potent, safe, and palliative treatment option [17]. In the context of neoplasm PDT, reactive oxygen species (ROS) play a pivotal role in orchestrating the destruction of tumor tissue [18,19,20]. Despite their crucial role, conventional PDT often exhibits limited cytotoxicity mediated by ROS. This limitation is primarily ascribed to the hypoxic state of the tumor microenvironment (TME) and the obstacles posed by the ROS defense mechanism, which is chiefly governed by the reductive agent GSH [21,22,23]. Consequently, therapeutic efficacy is compromised, and significant challenges persist in contemporary PDT techniques [24,25,26,27,28,29].

The hypoxic nature of the tumor environment can be attributed to the rapid proliferation of solid tumors and the consumption of oxygen during PDT [24,28]. To improve the therapeutic efficacy of traditional PDT, synergy therapies between PDT and various treatments have been widely reported [29,30]. Among them, the Fe^3+^ ion can be triggered to generate ROS by the Fenton reaction, which further exacerbates the intracellular hypoxia condition for the enhancement of photodynamic efficacy by amplifying the oxidative stress damage [31,32,33]. Recently, Pan et al. reported the preparation of nanoscale metal-organic frameworks (NMOF) containing 5,10,15,20-tetra(4-carboxyphenyl) iron porphyrin by dispersing in polyvinylpyrrolidone (PVP), which could be constructed as the nano-drug by enveloped NMOF and the hypoxia-activated tiapazine (TPZ) in the cancer cell membrane for tumor inhibition [34]. The presence of Fe^3+^ curbs oxidative resistance catalyzed by GSH and triggers the Fenton reaction, thereby generating ROS. This ROS production exacerbates hypoxia, activating the chemotherapeutic agent TPZ in conjunction with PDT to eliminate tumors. However, the challenge of the hypoxic TME remained unaddressed for synergy therapies of the two drugs. Shi et al. employed Fe^3+^-containing hybrid nanospheres, an aggregation-induced emission photosensitizer, and the Bcl-2 inhibitor sabutoclax, to fabricate nanoparticles via coordinately-driven self-assembly in an aqueous milieu. Fe^3+^ could induce the Fenton reaction to heighten ROS levels, while sabutoclax facilitated tumor apoptosis and mitigated inherent oxidative resistance by reducing GSH levels. This approach synergized with PDT to enhance therapeutic efficacy [35]. However, few investigations on the synergy therapy systems containing three drugs are reported, and their endogenous mechanisms associated with tumor hypoxia are lacking. 

HIF-1α, a hypoxia-inducible factor, is often upregulated to exhibit adaptive responses to the hypoxic condition of tumor cells [36]. In addition, the HIF-1α/vascular endothelial growth factor (VEGF) signaling cascade plays a pivotal role in coordinating tumor angiogenesis, cellular proliferation, and metastasis [37,38]. Based on the utilization of H_2_O_2_ enzymes, the degradation of the H_2_O_2_ concentration within tumors for improvement of hypoxia, which inhibits endogenous hypoxia-related pathways (e.g., HIF/VEGF hypoxia-related pathway), represents another approach to circumvent the limitations of PDT efficacy [39,40,41]. KC is a novel compound capable of inhibiting the translation of HIF-1α, demonstrating a potential antiangiogenic effect [42]. Consequently, KC is predominantly employed in the study of vascular diseases such as retinal vascular diseases, cardiovascular diseases, pulmonary hypertension, etc. To the best of our knowledge, few works have been reported on the utilization of KC to impede the HIF/VEGF pathway for alleviating tumor hypoxia endogenously and reversing oxidative resistance in tumors to enhance PDT efficacy [43,44].

In this work, novel carrier-free hybrid nanoparticles (KPF NPs) containing KC, PP, and Fe^3+^ ions were constructed via the supramolecular interaction. The morphology and photophysical properties of these KPF NPs were well characterized. Owing to its nanoscale size, KPF NPs could be endocytosed in the CAL-27 tumor tissues by the enhanced permeation and retention (EPR) effect. Applied as the new photosensitizer for PDT, KPF NPs showed the highest in vitro and in vivo therapeutic efficiencies for inhibiting the proliferation of CAL-27 cells under irradiation at a 650 nm laser compared to those of control formations, due to the efficient generation of ROS from PP and Fe^3+^ ion and the downregulation of the HIF-1α to restriction of ROS defense by reducing intracellular free radical scavenging. This work provides a new strategy to prepare a hybrid drug system for synergy therapy with PDT (Figure 1).

## 2. Materials and Methods

### 2.1. Materials

FeCl_3_ was purchased from Aladdin Reagents (Shanghai, China); Protoporphyrin IX, KC7F2, and DMSO were acquired from MedChemexpress (San Diego, CA, USA); 200-mesh carbon support films was sourced from Sigma Aldrich (Hamburg, Germany); CAL-27 cell line was obtained from ATCC (San Diego, CA, USA); 10 cm culture dishes, 6-well plates, and 96-well plates were procured from Sorfa (Shanghai, China); High-glucose DMEM, 0.25% EDTA, and CellROX^®^ kits (BD, San Diego, CA, USA) were bought from ThermoFisher (San Diego, CA, USA); 10% fetal bovine serum and 1% penicillin/streptomycin were purchased from Life Technologies (San Diego, CA, USA); PBS buffer was acquired from TBD (Shanghai, China); 4% paraformaldehyde fixative was purchased from Biosharp (Shanghai, China); SOSG, Hoechst 33,342 live cell staining solution (100×), MTT, Annexin V-FITC apoptosis detection kit, GSH and GSSG assay kits, Secondary antibodies for immunohistochemistry, DAB chromogenic reagent, anti-fluorescence quenching agent, and TUNEL apoptosis detection kit were acquired from Beyotime (China); RIPA lysis buffer, protein phosphatase inhibitor, BCA protein assay kit, 5× protein loading buffer, SDS-PAGE gel preparation kit, electrophoresis buffer, TBS buffer, skim milk powder, Tween 20, ECL luminous reagent kit, β-actin antibody, GADPH, Histone H3, HRP-labeled goat anti-rabbit secondary antibody, and HRP-labeled goat anti-mouse secondary antibody were purchased from Servicebio (Shanghai, China); 0.22 μm and 0.45 μm polyvinylidene fluoride(PVDF) membranes was bought from Millipore (USA); BALB/c mice were purchased from Chinese Academy of Sciences Laboratory Animal Center (Shanghai, China); Hematoxylin and Eosin (H&E) staining kit were bought from Solarbio (China); and HIF-1α and VEGF-A antibodies were acquired from Proteintech (China).

### 2.2. Synthesis of KPF NPs

In a typical case of synthesizing, FeCl_3_ (1.62 mg, 10 mmol) and ultrapure water (2 mL) were introduced into a 10-mL vial. The PP and KC solutions generated by dissolving PP (5.62 mg, 10 mmol) and KC (5.70 mg, 10 mmol) in 50 μL of DMSO were mixed and stirred. The mixing process was allowed for 4 h, followed by the preparation of KPF NPs. To complete this stage, the above-mixed solution was dropped into FeCl_3_ solution with stirring, which continued for another 24 h. In the sequential method, the obtained KPF NPs were purified by ultrafiltration centrifugation (10,000 rpm, 15 min) for the purpose of eliminating DMSO, free FeCl_3_, and free drugs. By adopting ultrasonic treatment during the conduct above, the reaction mixture was completely dissolved. The product was isolated from the reaction mixture in the form of an aqueous solution, which was lyophilized and turned into a powder.

### 2.3. Characterization of KPF NPs

Prior to analyzing the characteristics, nanoparticles were examined by a transmission electron microscope (TEM; Tecnai G2 Spirit Biotwin; FE-381 TEM, TALOS F200X). This specific type of microscope is set on the formvar/carbon-supported copper grids (Sigma Aldrich; size 200 mesh). The KPF NPs solution obtained from the previous stage was directly used for dynamic light scattering (DLS) analysis (omni, Brookhaven, San Diego, CA, USA), enabling a further study of size distribution. Since the iron concentrations were to be measured, a Thermo ICAP 6300 Duo View Spectrometer with a solid-state CID detector was applied, as an inductively coupled plasma optical emission spectrometer (ICP-OES) on it was contributive. The ultraviolet-visible (UV-Vis) spectra, on the other hand, were measured by a spectrophotometer (UV-1800, Shimadzu, Japan). In terms of FTIR spectra, data were obtained by use of a spectrophotometer (Nicolet 6700, Thermo Scientific, Waltham, MA, USA) in potassium bromide pellets. Moreover, X-ray photoelectron spectroscopy (XPS) measurements on freshly cleaved samples were performed using an ESCALAB250 spectrometer with an Al Kα source operating at 10 kV. The pressure in the analyzer was less than 6.67 × 10^−6^ Pa. Additional binding energy scans were carried out for Fe 2p core level regions. Hybrid KPF nanoparticles were confirmed for the distribution of carbon, oxygen, nitrogen, sulfur, and iron through scanning transmission electron microscopy (Thermo Fisher, San Diego, CA, USA). The ^1^O_2_ production of KPF added to the H_2_O_2_ (10 mM) solution was measured by SOSG under laser irradiation for 180 s. Fluorescence intensity values were detected using a plate reader (SpectraMax i3, San Diego, CA, Molecular Devices, USA). The content of Fe^3+^ ions in KPF NPs was determined by the ICP test (Avio 500, San Diego, CA, USA).

### 2.4. Cell Culture

CAL-27 squamous cell lines were selected for both in vitro and in vivo experiments. The cell lines were generously provided by ATCC. For optimal growth, CAL-27 cell cultures were established in Dulbecco’s Modified Eagle Medium (Life Technologies, San Diego, CA, USA), supplemented with 10% fetal bovine serum and 1% penicillin/streptomycin (Life Technologies). CAL-27 cells were incubated at 37 °C in an atmosphere of 5% CO_2_ and 21% O_2_ under normoxic conditions.

### 2.5. In Vitro Assessment of KPF NPs

The cellular uptake of the KPF NPs underwent an investigation in vitro by exposing CAL-27 cells to the particles. Specifically, CAL-27 cells were seeded in six-well plates at a density of 2 × 10^5^ cells per well in two milliliters of complete DMEM culture medium, waiting for a 24 h time period for culturing. Next, the KPF NPs, which dissolved in the DMEM culture medium, were added to separate wells. Meanwhile, the cells were incubated at 37 °C for predetermined time intervals, washed with PBS, and fixed with 4% paraformaldehyde for 30 min at room temperature. After that, the slides were then rinsed three times with cold PBS. Finally, the nuclei were stained with Hoechst 33,342 for 6 min, and the pretreated cells were observed under a confocal laser scanning microscope (CLSM) (LSM900, Zeiss, Shanghai, China). Technical abbreviations will be explained upon first use; the cellular uptake of KPF NPs evaluation will be undertaken at the cellular level with flow cytometry (BD, San Diego, CA, USA).

Also, to study intracellular ROS, flow cytometry was employed. Following drug exposure, cells were incubated at 37 °C for 15 min with CellROX^®^ oxidative stress agents (BD, San Diego, CA, USA) (2 μL/mL), washed with PBS, fixed with 4% paraformaldehyde for 15 min at room temperature, and rinsed 3 times with cold PBS. Levels of ROS were then assessed at the cellular level using flow cytometry and analyzed through a FACS Calibur instrument.

To elucidate the intracellular variations of GSH and the GSH/GSSG ratio, CAL-27 cells were seeded at a density of 2 × 10^5^ cells per well in six-well plates. These cells were subsequently treated with FeCl_3_, PP, KC, PP/KC mixture, and KPF nanoparticles. The subset designated for phototherapy underwent irradiation (100 mW/cm^2^, 5 min) and was then incubated for an additional 4 h. Subsequent to this, cells were processed employing GSH and GSSG assay kits in accordance with the manufacturer’s protocols. The absorbance values at 412 nm for each treatment group were ascertained utilizing a Bio Molecular Devices spectrophotometer. Based on standard curves, intracellular GSH concentrations and GSH/GSSG ratios were computed for each experimental cohort.

From another aspect, the cell cytotoxicity study demands a focus on CAL-27 cells. To achieve the goal, CAL-27 cells were seeded at a density of 10,000 cells per well in 200 μL medium, then incubated overnight in 96-well plates. Afterward, the cells were treated with different concentrations of PP, KC, KPF NPs, and FeCl_3_. We should point out that this must be undertaken in the presence or absence of light irradiation (100 mW/cm^2^, 5 min) and be incubated for 48 h. In the following step, 20 μL of 5 mg/mL MTT assay stock solution in PBS was added to each well, and the plates were incubated in the dark for 4 h. Then, with the formazan crystals produced, they were solubilized in 100 μL per well of DMSO. The absorbance of the solution in each well was measured at 490 nm using a Molecular Devices spectrophotometer. Each assay was repeated 3 times. In addition, cell apoptosis analyses were also verified using flow cytometry. The cells were incubated for 15 min with 5 μL of Annexin V-FITC and 10 μL of propidium iodide (PI) at 37 °C. The samples were then analyzed using a FACS Calibur instrument. To be precise, the FACS Calibur instrument was utilized to analyze the samples and determine the total number of cells experiencing apoptosis. 

As for determining HIF-1α and VEGF-A expression levels after treatment, CAL-27 cells were pretreated with PBS, PP, KC, PP/KC mixture, and KPF NPs at the same concentration (PP 4 μM, KC 8 μM). All of them were irradiated for 5 min. What was observed 24 h after irradiation was that cells were lysed, and the concentration of whole lysate protein was determined with BCA reagents. Moreover, 20 μg of protein was separated using a 12% SDS-polyacrylamide gel before it was wet-transferred to a poly (vinylidene fluoride) (PVDF) membrane. The membranes were blocked and incubated by primary antibodies at 4 °C overnight. Additionally, it demands horseradish peroxidase-conjugated secondary antibody for 1 h at room temperature. The protein bands were eventually visualized with an enhanced chemiluminescence (ECL) substrate kit and analyzed using Image J software. 

### 2.6. In Vivo Study of KPF

CAL-27-bearing nude female mice (*n* = 3) were intravenously injected via tail vein with 200 μL of KPF (PP: 5 mg/kg) to the subject mice. In contrast, mice without any treatment were used as negative controls. Fluorescence intensity was monitored while using an in vivo imaging system, PerkinElmer/*IVIS spectrum, with an appropriate wavelength (E_x_ = 575 nm, E_m_ = 630 nm).

When the volume of the tumor grew up to nearly 100 mm^3^, CAL-27 xenograft nude mice (*n* = 5) were injected PBS or KPF NPs (10 mg/kg) by vein with 530 nm laser irradiation (100 mW/cm^2^, 5 min) or under dark conditions once every 3 days for 21 days. We measure the tumor volume using the following formula:V (mm^3^) = 1/2 × length (mm) × width (mm)^2^(1)

After 21 days of treatment, the tumors were removed, photographed, weighed, and then fixed with 4% paraformaldehyde for histological analysis. The collected tumors were further used for H&E, immunohistochemistry, TUNEL immunofluorescence, and blood biochemistry study.

### 2.7. Results of Statistical Analysis

Cell experiments were designed independently. In three independent trials, cells were represented by MRI images using biological triplicates. It had to be noted that the outcomes are presented as the mean ± standard deviation unless stated otherwise. In that case, statistical analysis was carried out using an ordinary test with no matching or pairing to compare two or more groups with Tukey’s multiple comparison test, with the variance confined to a single pool. The level of significance was set at *p* < 0.05 in comparison to the control group. Prism 9.3.0 software (GraphPad) was used for this occasion. Protein grayscale in Western Blot experiments taken with AlphaEase FC software.

## 3. Results

### 3.1. Synthesis and Characterization of KPF NPs

To prepare the carrier-free hybrid nanoparticles, the mixture solution of PS of PP and HIF inhibitor of KC in DMSO was added into the FeCl_3_ aqueous solution under a gentle stirring condition at room temperature. After centrifugation for the removal of free drugs and DMSO, the ultrafine KPF NPs were collected and used by reprecipitation in water with the assistance of the ultrasound technique, and the yield was determined to be 70% with a mass of grams (Figure 1A). The size of KPF NPs was characterized by dynamic light scattering (DLS) analysis. In Figure 1B, the average diameter of hybrid KPF NPs is 47 ± 5 nm with a narrow PDI (0.195). The morphology of these NPs was further evaluated by TEM measurement. TEM illustrates that KPF nanoparticles maintain their spherical shape with a diameter of approximately 76 nm (Figure 1C), which is well consistent with that of the DLS result. As shown in Appendix A, the stability of KPF NPs was also demonstrated by the DLS technique. As discovered through SEM-EDX mapping of KPF NPs in Figure 1D, elements of carbon (C), nitrogen (N), oxygen (O), sulfur (S), and iron (Fe) are clearly found, suggesting the uniform distribution of three drugs in these nanoparticles through a traditional self-assembly approach. To verify the chemical state of Fe^3+^ in the KPF NPs, an XPS measurement was carried out. The XPS survey spectra of KPF NPs show the structural differentiation of C, N, O, S, and Fe elements (Appendix A). In Figure 2A, the high-resolution XPS of the Fe 2p spectrum exhibits two peaks at 710.5 and 723.9 eV, which could be attributed to the Fe 2p^3/2^ and 2p^1/2^ binding energies for Fe^3+^. Moreover, another two satellite peaks of Fe(III) are at 714.1 and 728.1 eV, respectively.

To identify the self-assembly driving force in KPF NPs, the UV-Vis spectra of compounds have been investigated (Appendix A). The maximum absorption with the peak at 440 nm in water is attributed to the strong short-wave Soret band of PP, originating from the π-π* transition of porphyrin backbones, while the weak Q-bands of π-π transition around 500–650 nm are also found. Compared with that of PP, the neglected red or blue shift of the UV-vis spectrum happened in KPF NPs. Moreover, the fluorescence emission profile of KPF NPs is similar to that of PP, suggesting the weak π−π stacking between PP and KC (Appendix A). Owing to the existence of amine, sulfonyl, and carboxyl groups in these drugs, temperature-dependent FTIR spectroscopy was employed to ascertain the presence of hydrogen bonds. In Figure 2B, the stretching vibration of C=O in PP is found at 1700 cm^−1^, which shifts to the higher wavenumber with the increasing temperature, demonstrating the dislocation of the hydrogen bond. 

For Fe^3+^ ions, it is easy to coordinate with heteroatoms like O, N, S, etc. The UV-Vis spectra of PP with different Fe^3+^ show hardly any change in comparison with that of bare PP (Appendix A). However, Figure 2C shows the apparent absorption peak at 250 nm and 295 nm when the Fe^3+^ ion solution is added to the KC. The intensity of this peak reaches a plateau when the molar ratio of Fe^3+^ and KC is 2:1, suggesting the coordination of Fe^3+^ with the S=O and N-H groups. Finally, the contents of PP and KC in the KPF NPs were calculated based on the concentration-absorption intensity standard curve from LC-MS spectra (Appendix A), which is 19.47 and 10.21 μM, while the content of Fe^3+^ ion was 139.297 mg/L, determined directly by ICP test. Furthermore, the SOSG intensity showed that the KPF NPs in the H_2_O_2_ solution produced more ^1^O_2_ than KPF NPs in PBS when exposed to irradiation and without irradiation, indicating the ability of Fe^3+^ ions to induce the Fenton reaction with H_2_O_2_ to synergistically enhance the efficiency of PDT (Figure 2D).

### 3.2. In Vitro Study of KPF NPs

Due to the low π-π stacking between PP and KC, KPF NPs show good emission properties. The in vitro cellular uptake of the KPF NPs was examined against CAL-27 cells by laser scanning confocal microscope. After incubation with KPF NPs, time-dependent intracellular fluorescent images were observed in Figure 3A. The red emission is attributed to the excited PP drug. With the increasing incubation time, the intensity of red fluorescence increases, demonstrating the successful internalization of KPF NPs by CAL-27 cells. 

Thereafter, CAL-27 cells were stained with an intracellular ROS indicator (2′,7′-dichlorodihydrofluorescein diacetate (DCFH-DA)), which can be used to evaluate intracellular ROS in CAL-27 from its green fluorescence. Based on the internalization of nanoparticles by CAL-27 cells, the prominent green fluorescence in CAL-27 cells pretreated with different drug groups was monitored by flow cytometry analysis (Figure 3B). In Appendix A, the intensity of fluorescence is slightly enhanced when the CAL-27 cell is pretreated with FeCl_3_ and KPF NPs without irradiation, in comparison with that of control, suggesting the ROS generation from Fe^3+^ ion via a Fenton reaction. Upon exposure to light, the bright fluorescence in CAL-27 cells pretreated with KPF NPs is achieved, indicating the considerable intracellular ^1^O_2_ generation potential from KPF NPs under light by excitation of PP. 

To understand the potential of KC for HIF-1α inhibitor in curbing tumor cell proliferation, the intracellular GSH content in CAL-27 cells was determined by GSH and GSSG assay kits. In Figure 3C and Appendix A, both FeCl_3_ and PP without KC show a similar GSH and GSSG content to that of the control. However, these administrations containing KC exhibit low GSH and GSSG content in CAL-27 cells, suggesting that HIF-1α inhibitor KC has a good effect on reversing the hypoxic microenvironment and GSH oxidative resistance. 

On the basis of high ROS generation and low GSH expression from KPF NPs, the in vitro MTT assays of various formulations, including KPF NPs and other controls, were performed on CAL-27 cells after incubation for 24 h. Without light illumination, all of PP, FeCl_3_, MIX, and KPF NPs show low cytotoxicity against CAL-27 cells (Appendix A). After pretreated KPF NPs with the 10 μM concertation of PP, the cell viability of CAL-27 cells is still higher than 70%, suggesting the relatively low dark cytotoxicity of PP and limited inhibition efficiency of FeCl_3_ and KC. Upon irradiation at 650 nm, a high level of intracellular ^1^O_2_ would be generated from PP; thus, these PP, PP-containing MIX, and PP-containing KPF NPs show good tumor inhibition performance (Figure 3D). The IC_50_ value of CAL-27 cells in the PP cohort was 8.24 ± 0.7 μM. The introduction of Fe^3+^ ion and KC into PP could enhance the cytotoxic responsiveness, resulting in the lower IC_50_ value of 6.83 ± 0.5 μM. After these three drugs form the nanoparticles, the highest tumor inhibition properties with IC_50_ of 2.55 ± 0.49 μM are achieved due to the higher synergistic therapeutic efficiency via the nano-drug technology. 

The apoptosis effect of our nanomaterial and other formulations on CAL-27 cells could be investigated by an Annexin V-FITC/PI double-staining assay and was determined by flow cytometry analysis. Cells without any drug were used as controls. After incubation with these formulations with the PP dose of 4 μM without laser irradiation, the percentages of apoptosis induction in CAL-27 cells by FeCl_3_, PP, MIX, and KPF NPs are 5.25%, 15.09%, 14.01%, and 16.59%, respectively (Appendix A), which is a bit higher than 2.90% of control formulations. Under irradiation with a 650 nm laser, the percentages of apoptotic cells in CAL-27 cells by KC, PP, MIX, and KPF NPs are 23.1%, 47.87%, 59.9%, and 72.73%, respectively (Figure 4A). Compared to the other drug formulations, KPF NPs cause the highest apoptosis rate in CAL-27 cells, which is in agreement with those of MTT results. 

To verify whether the HIF-1α inhibitor KC in KPF nanoparticles has the effect of reversing the hypoxic microenvironment and GSH oxidative resistance, the influence of all formulations on intracellular HIF-1α and VEGF expression was evaluated in vitro by Western blotting assay. As shown in Figure 4B–D, HIF-1α protein expression is downregulated slightly by KC, PP, and MIX compared with control. In contrast, the expression of HIF-1α is remarkably downregulated by KPF nanoparticles. The inhibition of VEGF is essential for regulating tumor angiogenesis. As expected, the expression of VEGF is inhibited by the KC-containing drug formulations. The formulation of KPF NPs shows the significant downregulation of VEGF expression compared to the free KC and PP. These data indicate that the nanodrug promotes the regulation of protein expression during tumor inhibition.

### 3.3. In Vivo Anti-Tumor Study of KPF NPs

Owing to the nanosize and good tumor inhibition performance of our materials, KPF NPs would be accumulated at the tumor site through enhanced permeability and retention (EPR) effect for high-efficiency in vivo anti-tumor therapy. Firstly, KPF NPs were injected into the tail vein of CAL-27-bearing nude mice together with an equivalent dose of PP (5 mg/kg, 200 μL). As depicted in Figure 5A and Appendix A, the emergence of the fluorescence signal is found within the initial 2 h, indicating that KPF NPs reach the tumor region. The fluorescence intensity at the tumor region achieves the highest accumulation 12 h after the injection (Figure 5B), and this fluorescence signal remains bright within 24 h, suggesting that the KPF NPs exhibit a robust EPR effect within the tumor microenvironment. 

To further elucidate the anti-tumor efficacy of KPF nanoparticles, CAL-27-bearing nude mice were used to evaluate the traditional post-PDT treatment. Saline administration was used as the control. As shown in Figure 5C,D, tumors in the control formulation expand from an approximate volume of 100 mm^3^ to 760 mm^3^, suggesting the irradiation of laser (100 mW/cm^2^, 5 min) has limited tumor inhibition therapeutic effect. A marked retardation in tumor growth is discerned in the PP, KC, and combined PP and KC cohorts, evidenced by their tumor volumes augmenting to about 400 mm^3^, 250 mm^3^, and 150 mm^3^, respectively. In vivo tumor growth in tumor-bearing mice administered with KC shows an inhibition rate of almost 50%, providing that KC inhibits tumor cell proliferation by inhibiting HIF-1α expression. Notably, tumors pretreated with the KPF formulation consistently exhibit the least volumetric progression with 95% tumor growth inhibition, underscoring the potential of KPF nanoparticles in efficaciously curtailing tumor expansion. Furthermore, without laser irradiation, tumors continue to grow and increase with the increasing injection time, demonstrating the predominant tumor growth inhibition after 21 days of treatment resulting from the ^1^O_2_ generation from PP (Appendix A). These results are in agree with those of in vitro studies. During the 21 days of treatment, a slight drop in body weight is found for the saline- and KC-pretreated formulations because of their negative treatment effects from the low therapeutic efficacy. However, concerning the treatment of KPF NPs, the obvious increase in body weight is more observed than those of PP and MIX pretreated formulations, which suggests a steady body growth after the effective therapy for 21 days (Figure 5E). 

The immunofluorescence and immunohistochemical analysis were carried out to evaluate the PDT efficacy of KPF NPs and other formulations on the CAL-27 tumor-bearing mice. Histological examination of hematoxylin and eosin (H&E)-stained tumor tissue after treatment with KPF NPs and various formulations are given in Figure 6 and Appendix A. In the tumors treated with saline, FeCl_3_, and other groups without irradiation, large and spindle-shaped nuclei were visible, showing rapid growth of tumor cells. However, nuclear shrinkage and fragmentation of tumor cellularity were found in tumor tissues pretreated with PP and MIX under irradiation. Furthermore, KPF NP pretreated tumors show a large necrotic area of nuclei and neutrophil infiltration, suggesting effective inhibition of tumors by KPF NPs via a PDT strategy. Correspondingly, TUNEL assays show the same change tendency of fluorescence intensity. The brightest green fluorescence is observed clearly in the tumor tissue after treatment with KPF NPs, indicating the most significant anti-tumor effect of the KPF NPs treatment group. For the HIF-1α expression, the in vivo hypoxia microenvironment in tumor tissue pretreated with various formulations was investigated by immunohistochemistry (IHC). Figure 6 exhibits that the tumor cells pretreated with saline show positive expression of HIF-1α. When treatment with the KC, a less negative expression of HIF-1α than that of control is observed due to its nature of hypoxia-inducing HIF-1α expression inside the tumor. As expected, free PP or MIX-treated tumor cells have downregulated the in vivo HIF-1α expression inside the tumor, which resulted from the high ^1^O_2_ generation by excitation of PP with laser irradiation. Owing to the enhanced accumulation in tumor tissue, the protein expression of HIF-1α in tumor cells pretreated with KPF NPs is lowest among all therapeutic groups. Hence, KPF NPs with the synergistic effect of ROS generation and HIF-1α inhibition exhibit superior in vitro and in vivo anti-tumor efficacy by the PDT method.

### 3.4. Biological Safety Study

To further identify the biological safety of KPF NPs, the in vivo organ tissues were measured by H&E-stained immunohistochemical analysis. In the normal tissue slices, the large and spindle-shaped nuclei in all of the organs are observed after pretreatment with formulations of saline and KPF NPs, indicating that the KPF nanoparticles have no obvious toxicity to major organs (Appendix A). 

In addition, the mice’s blood biochemical index does not show any obvious difference between administration with the KPF NPs and saline (Figure 7). The concentration of aminotransferase and phosphatase among the mice also has not significantly changed, confirming the KPF NPs’ harmless nature to the liver.

## 4. Discussion

In this work, we constructed a novel nanoparticle based on the photosensitizer of PP, the HIF-1α inhibitor of KC7F2, and cofactor of Fe^3+^ via supramolecular interactions (eg., hydrogen bond, coordinate bond) in the buffer solution. The nanoparticles are stable spheroids with a diameter of 76 nm. Instead of anti-tumor drugs modified by nontherapeutic drug carriers, our strategy can ensure a drug-loading efficacy of 100%. Furthermore, our study reveals that the KPF NPs can enter the cancer cells and effectively inhibit their proliferation both in vitro and in vivo in comparison with that of PP, KC7F2, or PP/KC7F2 mixture, respectively. In the group of KPF NPs, effective ROS generation is produced by the irradiation of PP and the Fe^3+^ ion via the Fenton reaction. In addition, KC7F2 counteracts the tumor’s hypoxia-associated survival mechanisms through the reduction of GSH and GSH/GSSG levels. Consequently, KPF NPs showed significantly better anti-tumor effects, originating from their highest ROS level. The results of Western Blot confirmed that KPF NPs significantly reduced the level of HIF-1α and VEGF to alleviate the hypoxia environment of the tumor and restrain angiogenesis, indicating that the KPF NPs not only reversed the oxidative resistance but also inhibited the HIF/VEGF hypoxia-related pathway. 

Overall, the excellent achievement in regard to the cytotoxicity of KPF NPs under light can be attributed to three factors as following: Firstly, KPF NPs with nanoscale size easily accumulate in tumor tissue via the EPR effect, which could enhance the usage efficiency of drug and reduce the dose of drug for minimization of toxicity to normal cells. Secondly, the photosensitizer of PP could efficiently generate reactive oxygen species; thus, KPF NPs provide a unique therapeutic effect for tumor inhibition. Thirdly, the Fe^3+^ ion in the KPF NPs could achieve sufficient ROS (·OH, O^2−^) in the tumor via the Fenton reaction; while KC can quench excess GSH to further increase ROS yield for reversal of the hypoxia-related survival pathway of tumors, resulting in the enhanced therapeutic effect. These results in this study provide a theoretical basis for addressing the challenge of the hypoxic environment of tumors treated with traditional PDT. Additionally, these findings offer important practical guidance and application prospects for innovatively applying PDT in OSCC therapy.

## 5. Conclusions

In conclusion, we elucidate a sophisticated strategy entailing a carrier-free supramolecular delivery system containing photosensitizer of PP, an HIF inhibitor of KC, and Fe^3+^ ions. The structure and physical properties of these nanoparticles were well characterized. Upon light exposure, KPF NPs exhibit a proclivity for generating singlet oxygen (^1^O_2_), which amplifies cellular damage with the synergistic effect of KC, serving to counteract the HIF/VEGF pathway for attenuating the tumor cell’s defensive capacity against ROS, and Fe^3+^ ions, thus exerting potent oxidative stress via the Fenton reaction. Our meticulously crafted supramolecular nanoparticle paradigm underscores significant potential in augmenting anti-tumor efficacy, which is substantiated through rigorous evaluations both in vitro and in vivo.

## Data Availability

Data is contained within the article or Appendix A.

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
