# Peer review of "Carrier-Free Hybrid Nanoparticles for Enhanced Photodynamic Therapy in Oral Carcinoma via Reversal of Hypoxia and Oxidative Resistance"

_pharmaceutics, 2024, doi:10.3390/pharmaceutics16091130_

Round 1

Reviewer 1 Report

Comments and Suggestions for Authors

-Through the introduction, it is not easy to follow the necessity of reersal of hypoia and photodynamic therapy. An scheme is required to address it.

-Flow cytometry diagrams for ros must be represented.

-For apoptosis, it is suggested to add a bar diagram showing early apoptosis, late apoptosis, and necrosis.

-For real-time imaging of mice after NP administration, please proide images of other mice, too. It hae been triplicate and please add the images of other two replicates as supplementary information.

Comments on the Quality of English Language

Quality of English is acceptable for publication.

Reviewer 2 Report

Comments and Suggestions for Authors

The paper entitled “Caarrier-Free Hybrid Nanoparticles for Enhanced Photodynamic Therapy in Oral Carcinoma via Reversal oHypoxia and Oxidative Resistance” by X. Li, Z. Li, J. Zhou, Y. Li, Q. Zhao, X. Yang, Y. Su, L. Shi, and L. Shen reports the novel approach of preparing self-assembling drug nanoparticles from the Protoporphyrin IX as a photosensitizer, a cystamine-based inhibitor of HIF-1α and iron (III) ions for enhanced photodynamic therapy. The authors provide also a characterization of the obtained nanoparticles and thoroughly explain the processes occurring during treatment. The manuscript is well-written and lacks major errors, although some disputable points did occur in the text. Therefore I suggest to implement some minor corrections before the paper is published:

1.      Please ensure that references are edited correctly, with space separating them from the preceding words.

2.      Page 2, line 60; page 10, line 383: does the word “orchestrating” match the context of the paragraphs?

3.      I suggest changing the word “America” regarding the country for “USA”.

4.      Page 4, line 169: Please ensure that the Manfacturer’s name is correct

5.      Page 4, line 176: “The cell lines were generously generated by ATCC”, do you mean that the cell lines were provided free-of-charge? In this case I suggest changing the word “generated” for “provided”, “given” or similar.

6.      Page  5, lines 206, 217: Please ensure that the device name is written correctly.

7.      Page 5, line 229: The abbreviation “PVDF” is explained here, although it appears earlier, in line 137, in the Materials section. Please consider explanation of the abbreviation when it occurs the first time.

8.      Page 10, lines 402-403: “he fluorescence intensity at the tumor region achieves the highest accumulation after a 12 h injection” – Do you mean 12 h after the injection or 12 h post-injection? Please clarify the sentence.

9.      Page 11, line 440: There is an error in the FeCl3 formula (capital I instead of lowercase l).

10.  In the description of Figure 7 (page 12, lines 464-466), I believe the descriptions of the plots are duplicated and/or confused: A is marked as blood urea nitrogen, whereas on the plot the Y-axis is labeled as ALP, and so on. Please adjust the description to the Y-axis labels.

Reviewer 3 Report

Comments and Suggestions for Authors

In this work, the authors have synthesized nanosized supramolecular complexes combining in their structure the photosensitizer protoporphyrin,  HIF-1α inhibitor - KS and iron ions Fe3+, which directly participate in the Fenton reaction. The obtained nanoparticles were characterized by a complex of modern methods - transmission electron microscopy, dynamic light scattering, XPS-spectroscopy, etc. The antitumor activity of the obtained nanoparticles was tested under in vitro and in vivo conditions. It was found that the obtained nanoparticles have a pronounced antitumor activity in the presence of laser radiation, the expression of which far exceeds the activity of individual components of the obtained nanoparticles (PP, KC and Fe3+) and can be considered as a potential effective agents for photodynamic therapy of cancer. The relevance of work in this direction remains high due to the high prevalence of cancer. The authors have done a lot of experimental work. The presentation of the results in the manuscript is logical. The graphical material is done at a good level. Nevertheless, in the presented form this manuscript requires revision in accordance with the following remarks:

1. the authors should give a clear description of the nanoparticles they have obtained. They should give the expected structure. Ideally, it would be good to provide crystallographic data.

2. In the experimental part, the authors should explain the form in which the nanoparticles are isolated and the yield of this synthesis.

3. The authors should explain the choice of PP/KC/FeCl3 molar ratio for the synthesis of nanoparticles.

4. It is absolutely necessary addition by the authors to explain the mechanism of formation of nanoparticles and their stabilization. At the expense of what nanoparticles keep their dimensionality at the nanoscale?

5. Does the nanoscale dimensionality of the resulting particles persist with time?

6. What is the nature of the particles visualized on TEM? 

7. The data presented in Figure 2b indicating a band shift and decrease in band intensity upon heating should be rechecked using an approach that excludes sample preparation with KBr, due to the presence of adsorbed water in it.

8. Data on the ability of nanoparticles to generate singlet oxygen should be provided in comparison with data for PP, KC and FeCl3 to determine the efficiency of the nanoparticles obtained by the authors.

Comments on the Quality of English Language

The text of the manuscript needs to be checked by a native English speaker. Some sentences are difficult to understand.

Reviewer 4 Report

Comments and Suggestions for Authors

The submitted manuscript describes a preparation and an evaluation (in vitro and in vivo) of novel carrier-free hybrid nanoparticles (KPF NPs). Although all experiments have conducted well and the contents of the manuscript is interesting, there are insufficient explanations for some experimental results. If the authors revise the manuscript properly, the reviewer thinks that the submitted manuscript would be suitable for publication for “pharmaceuticals”.

The name of authors cited in the main text should be Regular, not Italic. (lane 72 and 80 and others) CM (lane 76), AIE (lane 81), and PS (lane 81) are not required. VEGF in lane 90 should be spelled out. If the authors want to use the abbreviation of “KPF NPs”, the phrase “containing PP, KC, and Fe3+ ions” should change to “containing KC, PP, and Fe3+ ions”. The resolution of Figure 1 should be improved. Where is KC7F2 (green) after construction of the nanoparticles? What is PDI? (lane 266) The presentation of data shown in Figure 1 and Figure 2 is not proper. The reviewer cannot understand how to sort experimental results including supplemental data. What is a basis that KPF NPs possess a diameter of approximately 76 nm”? Judging from the TEM image, the reviewer thinks the particle size of prepared KPF NPs would be less than 50 nm. What is BF in Figure 1D? The graph of Figure 2A is too difficult to see. The arrangement of numbers in the graph should be reconsidered. What does the change in FT-IR spectrum by temperature increase mean? The data of FT-IR are required in this research? Figure 2C and 2D should be rebuilt. It is difficult to see the graph because of the use of many colors. What is SOSG? Although the authors show the relative amounts of PP, KC, and Fe3+, there is no showing on the absolute amounts of PP, KC, and Fe3+ in prepared KPF NPs. The colorful presentation in Figure 3B, 3C, and 3D is not undesirable. The reviewer thinks that the presentation of Figure 4C and 4D is not required. The colorful presentation in Figure 5D and 5E is not undesirable. What do 5 tumor tissue samples shown in Figure 5B mean? The figure legend of Figure 7 should be reconsidered.

Comments on the Quality of English Language

Minor English proofreading is required.

Round 2

Reviewer 2 Report

Comments and Suggestions for Authors

Paper can be accepted

Reviewer 3 Report

Comments and Suggestions for Authors

The authors submitted the manuscript revised in accordance with the comments and suggestions of the reviewers. However, unfortunately, some of the comments were ignored by the authors. In particular, there is no information about the characterization of the isolated sample. The authors should indicate in what form they isolated the product from the reaction mixture and what it is - a crystal? Powder? Aqueous solution? What is the yield of the obtained product (in % and grams)?
